# Transcriptome profiling helps to elucidate the mechanisms of ripening and epidermal senescence in passion fruit (*Passiflora edulia* Sims)

**Changbao Li[1,2], Ming Xin[1,2]\*, Li Li[1,2], Xuemei He[1,2], Guoming Liu[1,2], Jiemin Li[1,2], Jinfeng Sheng[1,2], Jian Sun[1,2]\***

**1** Institute of Agricultural Products Processing, Guangxi Academy of Agricultural Sciences, Guangxi, Nangning, China, **2** Guangxi Key Laboratory of New Technologies for Storage and Processing of Fruits and Vegetables, Guangxi, Nanning, China

\* jiansun@gxaas.net (JS); xinming@gxaas.net (MX)

**Data Availability Statement:** All relevant data are within the paper and its supporting information files.

## Abstract

Passion fruit (*Passiflora edulia* Sims), an important tropical and subtropical species, is classified as a respiration climacteric fruit, and its quality deteriorates rapidly after harvest. To elucidate the mechanisms involved in ripening and rapid fruit senescence, phytochemical characteristic analysis and RNA sequencing were performed in purple passion fruit with different treatments, that is, 1-methylcyclopropene (1-MCP) and preservative film (PF). Comprehensive functional annotation and KEGG enrichment analysis showed that starch and sucrose metabolism, plant hormone signal transduction, phenylpropanoid biosynthesis, flavonoid biosynthesis, and carotenoid biosynthesis were involved in fruit ripening. Treatment with PF and 1-MCP significantly affected the transcription levels of passion fruit during postharvest storage. A large number of differentially expressed unigenes (DEGs) were identified as significantly enriched in starch and sucrose metabolism, plant hormone signal transduction and phenylpropanoid biosynthesis at the postharvest stage. The PF and 1-MCP treatments increased superoxide dismutase (SOD), catalase (CAT) and peroxidase (POD) gene expression levels and enzyme activities, accelerated lignin accumulation, and decreased β-galactosidase (β-Gal), polygalacturonase (PG) and cellulose activities and gene expression levels to delay cell wall degradation during fruit senescence. The RNA sequencing data for cell wall metabolism and hormone signal transduction pathway-related unigenes were verified by RT-qPCR. The results of this study indicate that the cell wall metabolism and hormone signaling pathways are closely related to passion fruit ripening. PF and 1-MCP treatment might inhibit ethylene signaling and regulate cell wall metabolism pathways to inhibit cell wall degradation. Our results demonstrate the involvement of ripening- and senescence-related networks in passion fruit ripening and may establish a foundation for future research investigating the effects of PF and 1-MCP treatment on fruit ripening.

**Funding:** This work was supported by a grant from the Major Program of Science and Technology in Guangxi (Guike AA17204038), the Key Research and Development Program of Guangxi (Guike AB18221110, Guike AB18294027), the Special Fund for the 'Bagui Scholars' of Guangxi ([2016] 21), and Basal Research Fund Project of Guangxi Academy of Agricultural Sciences (Guinongke 2018YT26).

**Competing interests:** The authors have declared that no competing interests exist.

## Introduction

Passion fruit (*Passiflora edulia* Sims) is an important tropical and subtropical species with high commercial value due to its attractive flavor and aroma and nutrients with beneficial properties for human health, such as vitamins C, B1 and B2 and essential amino acids, as well as various minerals and fibers [1, 2]. This species can be planted and grown in many regions and is in high demand in the market for fresh fruit and processed foods and juices [3]. Passion fruit has been reported to have high contents of phytochemicals with antioxidant, anti-inflammatory and anticancer properties, which are associated with a lower risk of cardiovascular diseases, cancer and metabolic diseases [4, 5].

Fruit ripening and senescence, which are unique to plants, are complex biological processes. According to the ripening and senescence processes, fruit are divided into two groups, climacteric and nonclimacteric fruit [6]. Passion fruit quality deteriorates rapidly after harvest, and this fruit is classified as climacteric fruit, as manifested by a high respiration rate, pericarp shriveling, weight loss, flavor changes and pathogen increases, which can cause considerable economic losses and significantly restrict the supply chain of commodities [7]. During fruit development, ripening is considered a functionally modified form of senescence associated with ROS accumulation, respiration rate, ethylene production rate, and phytohormone level. Therefore, mature fruit with high antioxidant enzyme activity may present a prolonged fruit storage life and maintain quality attributes for longer periods [8, 9].

In recent years, a large number of studies have reported the postharvest conservation of various fresh fruits. The method of applying chemical or physical preservatives is widely employed to conserve fresh fruit because this method is the simplest and least expensive [3]. For climacteric fruit, the ethylene production rate usually increases rapidly during storage, which may accelerate the process of fruit softening or lead to physiological disorders [10]. Previous studies reported that the postharvest treatment with 1-methylcyclopropene (1-MCP) can effectively delay fruit ripening and softening processes and prolong the storage life of various fruits [11, 12] because it can bind to ethylene receptors in plant cells and inhibit ethylene action of fruit by suppressing ethylene biosynthesis-related genes expression levels [13, 14]. The effect of 1-MCP treatment on postharvest fruits was involved in many parameters, including fruit cultivars [15], concentration [16] and exposure time [17]. To reduce the addition of chemicals to fruit or food, preservative film (PF) provides an effective method for storing fruit after harvest. However, few studies have investigated the postharvest storage of passion fruit by the application of 1-MCP or PF. To date, no study has reported the suppression of passion fruit senescence and softening during poststorage treatment of PF and 1-MCP.

Although a number of studies have reported a reduction in the postharvest conservation of passion fruit by low temperature [3, 18], few concrete studies have revealed the mechanism and factor-associated signaling pathways involved in the ripening and senescence processes. To better understand the mechanisms involved in ripening and rapid fruit senescence, in the present study, phytochemical characteristics and RNA sequencing were conducted in purple passion fruit with different treatments (1-MCP and PF). Our results may provide a reference and baseline information to assist in the development of ripening and postharvest management protocols for passion fruit.

## Materials and methods

### Plant sample collection and treatment

Passion fruits (*Passiflora edulia* Sims), 'Tainong1', were obtained from orchards at the Guangxi Academy of Agricultural Sciences, China. During the process of ripening, the pericarp color of

passion fruit was green to purple. Samples were collected for different growth stages and different treatments during ripening storage and were divided into 9 groups (e.g., A, B, C, D, E, F, G, H and J). The A, B and C groups were collected at 4 weeks, 6 weeks and 8 weeks (mature stage) after anthesis (WPA), respectively. After harvest, the D and E groups were stored under ambient temperature conditions (20−25 ˚C and 75−80% relative humidity) at 4 d and 8 d, respectively. The F and G groups were treated with PF at 4 d and 8 d after ripening, respectively. The H and J groups were treated with 0.5 μl/L 1-MCP at 4 d and 8 d after ripening, respectively. All samples were stored at room temperature with 20−25 ˚C and 75−80% relative humidity. Three biological replicates were employed for each time point. Each peel (pericarp) of passion fruit was surface-sterilized with 3% $H_2O_2$ for 10 mins and vigorously rinsed with distilled water (>200 ml/per time) 5 times. After being ground to powder in liquid nitrogen, the peel sample was stored at −80 ˚C for further analysis.

## Physiological index determination of passion fruit

Certain important physiological indexes of ripening and senescence in passion fruit, including membrane permeability (relative electrical conductivity), respiration rate and weight loss (shrinkage rate), were measured in the present study. Fifty passion fruits were selected to determine the relative electrical conductivity (REC), respiration rate and weight loss rate. The REC was measured using a conductivity meter following Wang et al.'s method [19]. The respiration rate was determined according to the method of Yang et al. [20]. We used a TEL-7001 Infrared $CO_2$ Analyzer (Telaire, Goleta, CA, USA) to determine the respiration rate of passion fruit and expressed it as mg CO2·kg$^{-1}$ fresh weight (FW)·h$^{-1}$. The weight of each fruit was recorded at 0 d and 10 d. The weight loss rate was determined as follows: (initial weight before storage-weight after storage)/initial weight before storage×100%.

## Antioxidant- and cell wall-related enzyme activity determination

In the present study, we determined the activities of antioxidant enzymes during ripening and senescence in each treatment group. These antioxidant enzymes, including superoxide dismutase (SOD), catalase (CAT) and peroxidase (POD), were tested in our study. The extractions and activities of SOD, CAT and POD were determined according to Chen et al. and Liu et al.'s method [21, 22] with several modification. Frozen pericarp tissue of passion fruit was homogenized, added to 5 ml of 80% (v/v) ethanol, and then centrifuged at 12,000×g for 10 min at 4 ˚C, using the supernatant to determine antioxidant capacity. SOD, CAT and POD activities were measured by the absorbance at 560 nm, 240 nm and 470 nm, respectively.

The activities of β-galactosidase (β-Gal), polygalacturonase (PG) and cellulase were measured in passion fruit at different time points. The crude enzymes were extracted according to the method described by Fan et al. [23] and Chen et al. [24] with several modification to determine the activities of β-Gal, PG, and cellulase. The results were expressed as U/kg.

## RNA extraction and RNA sequencing analysis

Total RNA was extracted from pericarp tissues of passion fruit using a mirVana miRNA Isolation Kit (Ambion) according to the manufacturer's instructions. A, B, C, D, E, F, G, H and J groups of the control and PF- or 1-MCP-treated samples of passion fruit were used for RNA library preparation and sequencing. First-strand cDNA synthesis was performed using random oligonucleotides and SuperScript II. Second-strand cDNA synthesis was performed using DNA polymerase I and ribonuclease H. Then, the cDNA libraries were sequenced on an Illumina sequencing platform (HiSeqTM 2500 or Illumina HiSeq X Ten). Three biological replicate samples from each time point were used for library construction, and each library was

sequenced once. The raw data containing adaptors and poly-N and low-quality reads were removed. Then, the sequence duplication levels of the clean reads were assembled into expressed sequence tag clusters (contigs) and *de novo*-assembled into transcripts, and the Q20 and GC contents were calculated by using Trinity [25]. Blastx (E-value < 0.00001) was employed to search for homologs of our assembled unigenes and annotated in protein databases, including NR, KOG, SwissProt and PFAM and the database. The best results were employed to determine the sequence orientations of the unigenes. The functional annotation by GO terms (http://www.geneontology.org) was analyzed using the program Blast2GO. The COG and KEGG pathway annotations were performed using Blastall software against the COG and KEGG databases, respectively [26].

## Real-time qRT-PCR verification

To verify the usability of the transcriptomic data, the relative transcript levels of 18 genes that were either significantly up- or downregulated were determined using quantitative real-time reverse-transcription PCR (qRT-PCR) (S1 Table). These genes included signal transduction, coloration, cell wall function, respiration and energy. Total RNA was reverse-transcribed to obtain first strand cDNA using the RevertAid First Strand cDNA Synthesis Kit (Fermentas, Lithuania) according to the manufacturer's instructions. Gene-specific primer pairs were designed using Primer 3.0. qRT-PCR was performed using the SYBR$^{\text{®}}$ Green PCR kit (Qiagen, 204054). Plant GAPDH was used as the internal reference to normalize the cDNA content. All genes were repeated 3 times. The mRNA expression level of genes was calculated using the $2^{-\Delta\Delta Ct}$ method.

## Statistical analysis

Three biological replicates were used for RNA sequencing and qRT-PCR analysis. All data for each passion fruit sample were statistically analyzed using Student's t-test (P< 0.05). Differentially expressed unigenes were defined as unigenes with FDR < 0.001 and fold change > 2. P-values < 0.05 were considered to be significant when identifying enriched GO terms and enriched KEGG pathways.

# Results

## Effect of PF and 1-MCP treatment on the physiological biochemical index of passion fruit

The relative electrical conductivity (REC) in the CK, 1-MCP- and PF-treated passion fruit increased on days 0D-10D (Fig 1a). However, the rate of increase in REC was significantly slower in the PF-treated group than in the 1-MCP-treated group at 2D-10D. The respiratory rate increased in both PF-, 1-MCP-treated and control passion fruit at 0D-10D. The respiratory rates of passion fruit treated with 1-MCP and PF were significantly lower than those of the CK group at 4D, 8D and 10D (Fig 1b). The weight loss (shrinkage rate) of the control and 1-MCP-treated passion fruit exhibited a rapid increase from 0D to 10D, while it was notably suppressed by PF treatment (Fig 1c). The control group showed a notably higher shrinkage rate than the 1-MCP- and PF-treated passion fruit following storage for 2D-10D. The membrane permeability, respiratory rate and weight loss in PF-treated fruit were significantly lower than those in the 1-MCP and CK groups at 2D-10D.

We determined ROS-scavenging enzyme activities, including SOD, POD and CAT. SOD activity increased during 0D-2D of storage, exhibiting a slow decline from 4D in the 1-MCP-treated group and 6D in the CK group followed by a sharp increase in the later stages at 8D

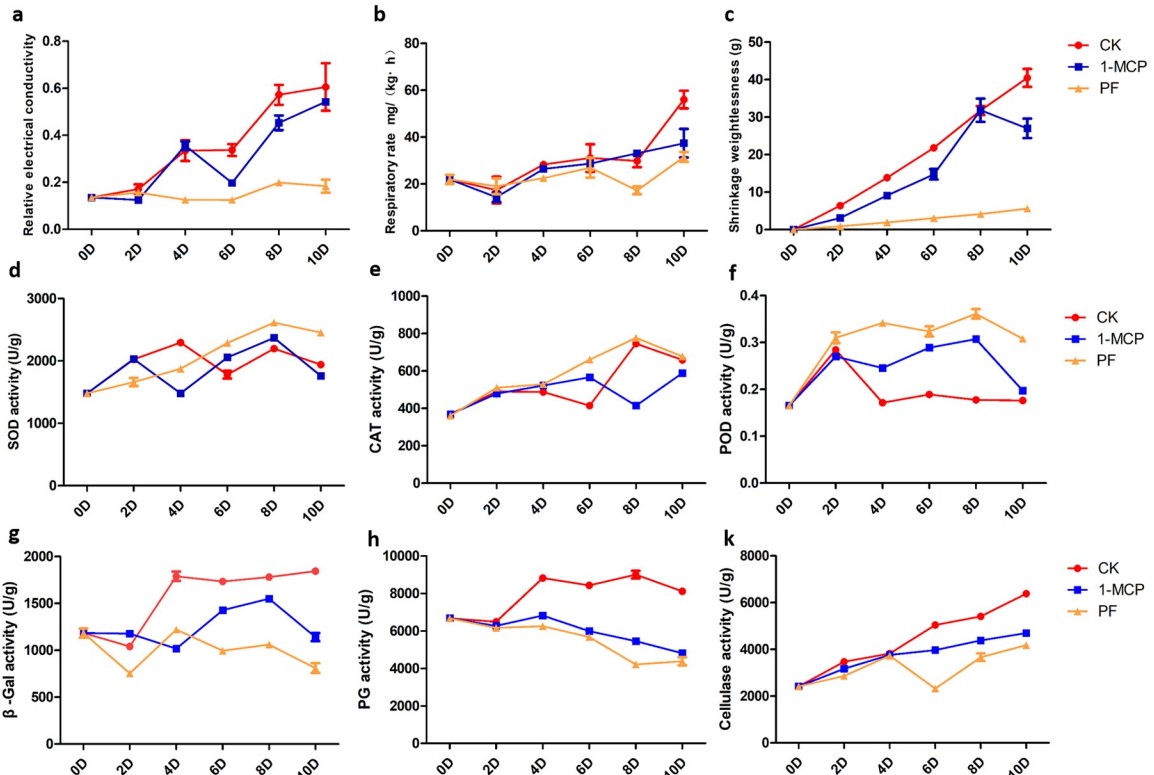

**Fig 1. The relative electrical conductivity (REC), respiration rate and weight loss rate (a, b and c) of passion fruit with 1-MCP and PF treatment during storage stages at room temperature (22–25˚C).** Antioxidant enzymes activities of superoxide dismutase (SOD), catalase (CAT) and peroxidase (POD) (d, e and f), and activities of β-galactosidase (β-Gal), polygalacturonase (PG) and cellulase (g, h and k) in control, 1-MCP and PF-treated passion fruit during storage stages at room temperature (22–25˚C). Values are the means ± SD.

and a slow decline from 10D (Fig 1d). These results showed that PF treatment activated SOD activity compared with 1-MCP and CK from 6D-10D. The CAT activity was higher in PF-treated fruit than in the 1-MCP and control groups (Fig 1e). For PF treatment, the POD activity was also higher in the PF-treated fruit than in 1-MCP-treated and control fruit (Fig 1f).

At all storage time points, the activity of β-Gal was significantly lower in PF-treated fruit than in the 1-MCP-treated and control fruit (Fig 1g). The activity of PG declined in both PF- and 1-MCP-treated passion fruit at 0D-10D. However, PG activity was higher in the control fruit than in the PF- and 1-MCP-treated groups. The cellulase activity of passion fruit was increased in both the 1-MCP- and PF-treated groups (Fig 1h). In PF-treated fruit, cellulase activity was significantly lower than that in the 1-MCP-treated and control groups at 4D-10D (Fig 1k). These results indicated that PF and 1-MCP treatment can significantly delay the degradation of cell wall components and postpone the senescence of passion fruit during postharvest storage by reducing β-Gal, PG and cellulose activities.

## RNA-seq data analysis and functional annotation of *Passiflora edulia* Sims

Transcriptome sequencing was conducted to analyze the underlying mechanisms of maturation and senescence in passion fruit by 1-MCP and PF treatment. All stages of these samples were renamed A (4 weeks post-anthesis), B (6 weeks post-anthesis), C (8 weeks post-anthesis—physiological maturity), D (stored at room temperature for 4 d), E (stored at room temperature for 8 d), F (PF treatment for 4 d), G (PF treatment for 8 d), H (1-MCP treatment for 4 d)

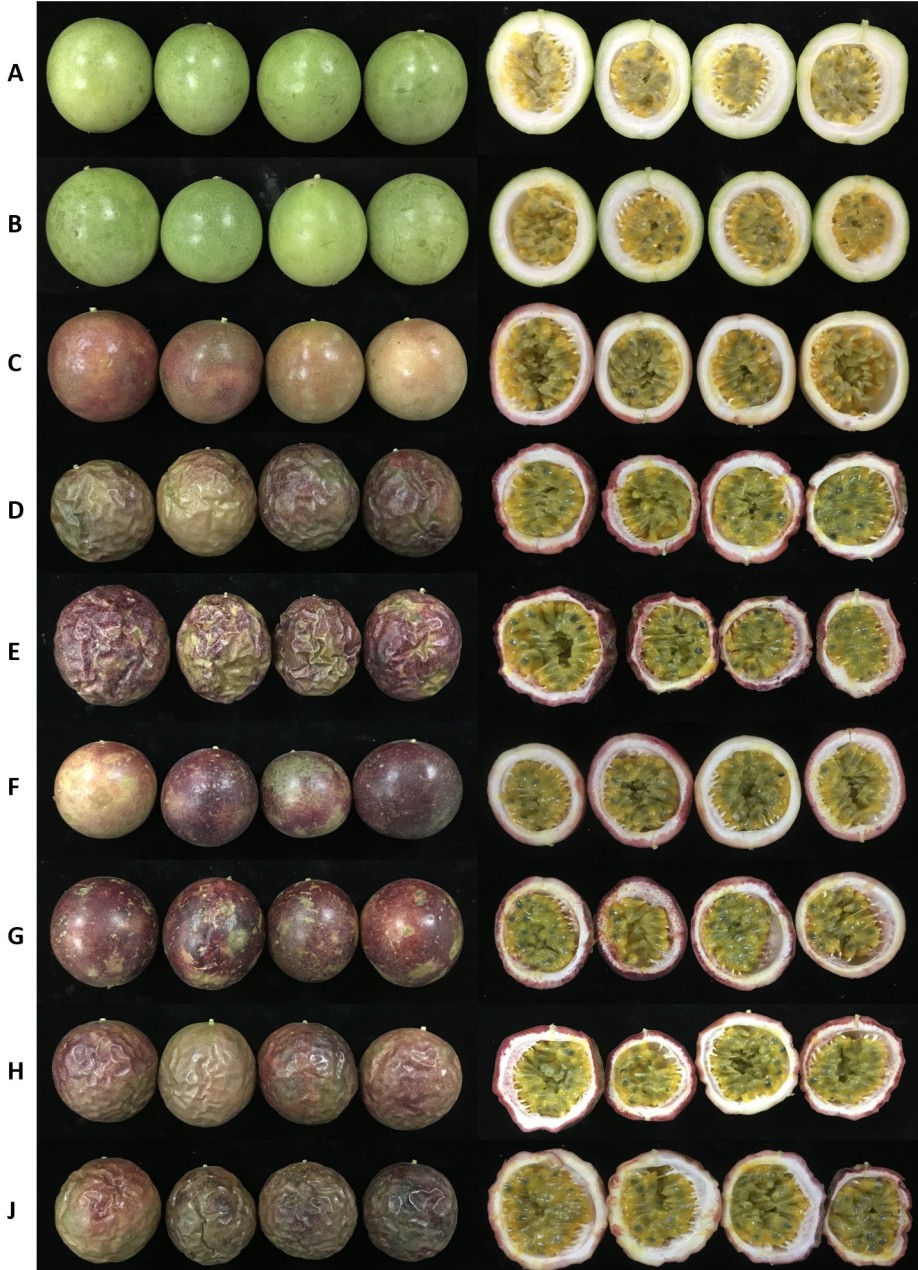

**Fig 2.** Phenotype images of 'Tainong1' purple passion fruits during ripening stage and effects of PF and 1-MCP treatments on the passion fruit at postharvest storage, including a (4 weeks post anthesis), b (6 weeks post anthesis), c (8 weeks post anthesis—physiological maturity), d (store at room temperature for 4 days), e (store at room temperature for 8 days), f (PF treatment for 4 days), g (PF treatment for 8 days), h (1-MCP treatment for 4 days) and j (1-MCP treatment for 8 days).

and J (1-MCP treatment for 8 d) (Fig 2). The Q30 values of these samples were 92.85%-94.65% and 8.12 G-9.22 G raw reads were produced in different stages of passion fruit. The average GC content was 50.36%, and a total of 56628 unigenes were detected in all samples after removal of low-quality reads. The average length of these unigenes was 854 bp. These unigenes were annotated in seven databases. Among these genes, 54.87% were annotated in GO, 23.01%

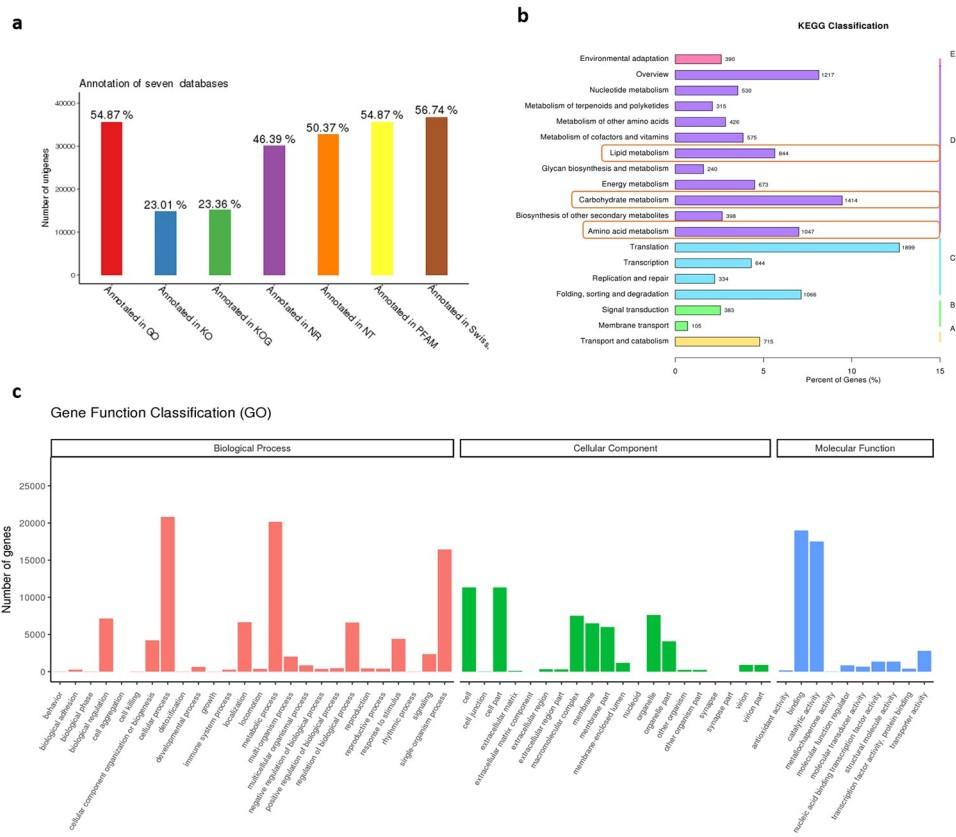

**Fig 3.** The number of unigenes annotation in GO, KO, KOG, NR, NT, PFAM and Swiss-port database by RNA_seq data analysis (a). KEGG classification (b) and Gene ontology (GO) functional classifications (c) of all expressed genes detected by RNA-Seq in passion fruit at all stages (ripening and storage stage).

were annotated in KO, 23.36% were annotated in KOG, 46.39% were annotated in NR, 50.37% were annotated in NT, 54.87% were annotated in PFAM and 56.74% were annotated in Swiss-Prot (Fig 3a). The results of GO and KEGG annotation are shown in Fig 3. We found that lipid metabolism, carbohydrate metabolism and amino acid metabolism were enriched in KEGG (Fig 3b). Metabolic processes, single-organism processes and catalytic activity were enriched in GO (Fig 3d).

## Expression analysis of RNA-seq data in *Passiflora edulia Sims*

Principal component analysis (PCA) showed that the first principal component (PC1) could explain 59.36% of the total variance and distinguish samples based on the time of storage at 8D (E) from other groups. The E group was significantly different from the other groups. At the 8D stage, the J group (1-MCP treatment) was closely related to the E group compared with the G group. The G group (PF treatment) was similar to the C and D groups. Then, the second principal component (PC2) explained 12.4% of the total variance and separated different stages of passion fruit according to the 5 time points (S1a Fig). The results suggest that 1-MCP and PF treatment may delay the senescence of passion fruit, and the effectiveness of the PF-treated group was better than that of the 1-MCP-treated group.

The DEGs were identified in the A-J group of passion fruit by comparing the FPKM values of each unigene based on the criteria log2 (fold change) $\geq$ 2 and significance p < 0.005. The

result was consistent with the PCA results. The expression levels of DEGs in the E group (CK for 8D) were notably differentially expressed from those in the G (PF treatment for 8D) and J (1-MCP treatment for 8D) groups. However, the expression levels of DEGs in the D group (CK for 4D) were not significantly different from those in the F (PF treatment for 4D) and H (1-MCP treatment for 4D) groups (S1b Fig). Therefore, we speculated that 8D was a key stage in studying the underlying mechanism of senescence in passion fruit with PF and 1-MCP treatment.

## KEGG pathway analysis of DEGs during the ripening stage

First, we analyzed the DEG expression pattern during the mature stage of passion fruit at A, B and C. Significant differences in DEGs between the B and A groups and between the C and A groups were observed for 3189 and 6622, respectively (Fig 4a). Among the 9811 DEGs, 2481 were identified in both the B vs A and C vs A comparisons. Therefore, 708 and 4141 unique DEGs were expressed in the C vs A and B vs A comparisons, respectively (Fig 4a). The 2481 DEGs might be involved in the process of maturation in passion fruit. To analyze the biological pathways of DEGs, we employed these DEGs for Kyoto Encyclopedia of Genes and Genomes (KEGG) annotations. Then, we further employed these DEGs for KEGG pathway analysis. The top 20 KEGG pathways with the most significant enrichment are shown in Fig 4b. Among all KEGG pathways, most of these DEGs were significantly enriched in 'starch and sucrose metabolism' (ko00500), 'plant hormone signal transduction' (ko04075), 'phenylpropanoid biosynthesis' (ko00940), 'flavonoid biosynthesis' (ko00941), 'carotenoid biosynthesis' (ko00906), 'biosynthesis of unsaturated fatty acids' (ko01040), and 'diterpenoid biosynthesis' (ko00904). The results indicated that these pathways mainly participate in the maturation of passion fruit. With the process of passion fruit development, the color of the peel changed significantly (Fig 4) in the C group. Among these pathways, 'phenylpropanoid biosynthesis' (ko00940), 'flavonoid biosynthesis' (ko00941) and 'carotenoid biosynthesis' (ko00906) were related to fruit coloration. Therefore, we further analyzed the expression level of DEGs in these two pathways to explore the potential mechanism of coloration in passion fruit during maturation.

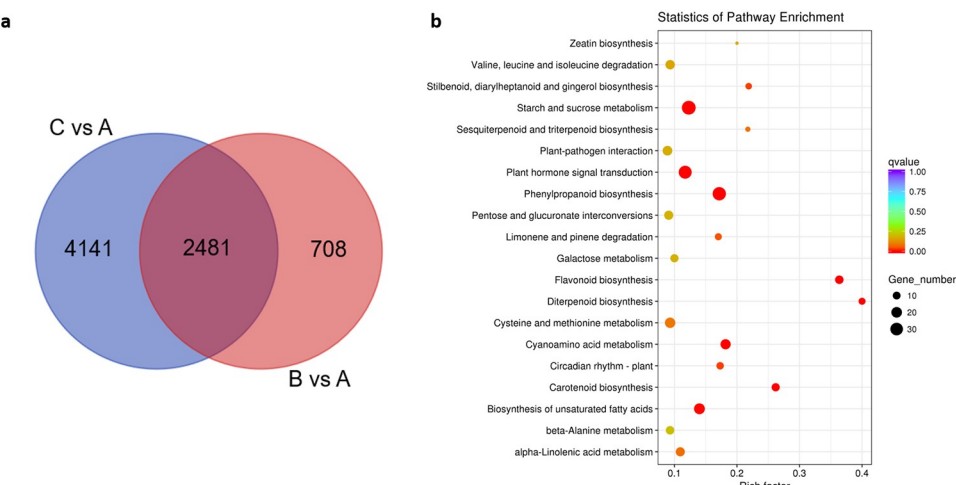

**Fig 4. The expression pattern of Differentially Expressed Genes (DEGs) and KEGG enriched pathway in purple passion fruits during ripening stage.** (a) Venn diagram of DEGs in C vs A and B vs A comparison groups. (b) Top 20 KEGG enrichment pathways in passion fruit during ripening stage. Rich factor was represent the degree of DEGs enrichment in this pathway.

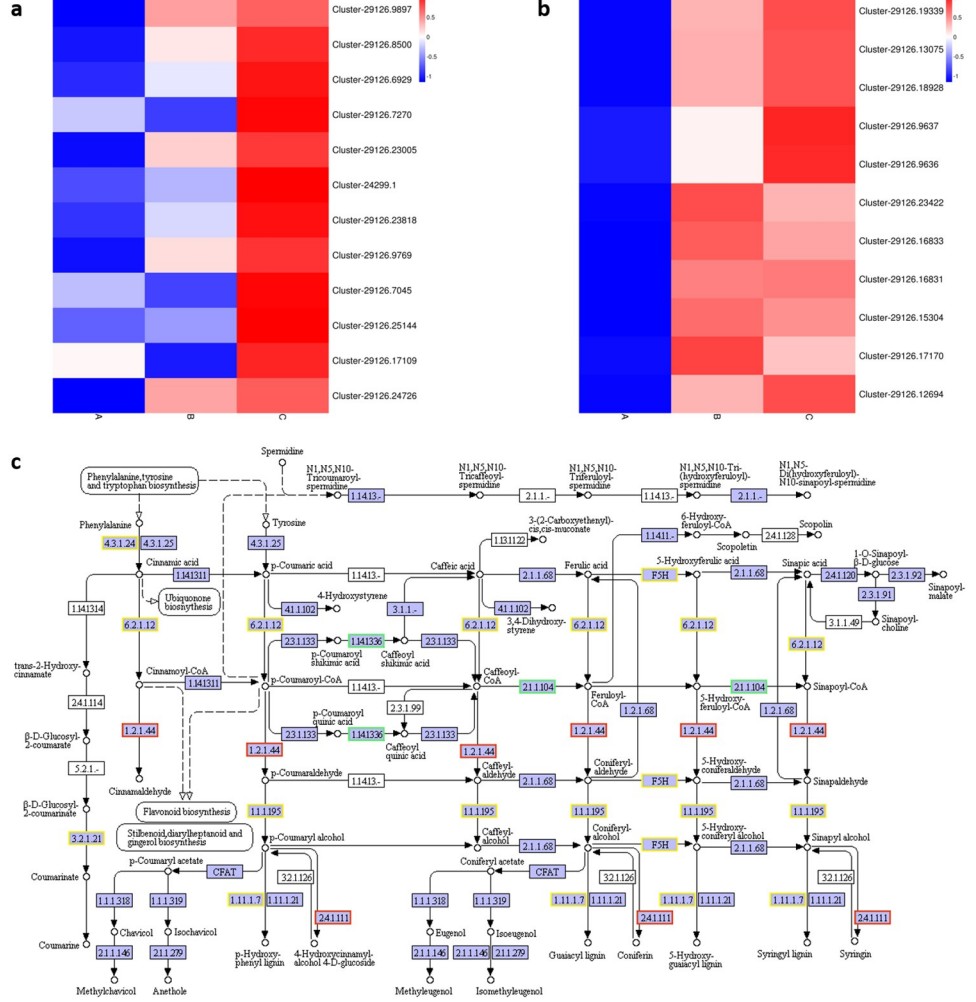

**Fig 5. Expression profile of DEGs expression related to flavonoids and anthocyanin synthesis (a), synthesis of carotenoids (b) and phenylpropanoid biosynthesis pathway (c).** Red indicates up-regulated and blue was down-regulated DEGs expression in heatmap. Red border in Fig 5c indicates up-regulated DEGs, green border was down-regulated DEGs and yellow border was both up- and down-regulated DEGs. The corresponding index are listed in SF1 and SF2 in S1 File. Data are presented as the mean± SD of three biological replicates.

## Pericarp color changes during the ripening stage

A total of 12 DEGs were identified to regulate 'flavonoid biosynthesis' (ko00941) in the present study. Compared with the A group, these 12 DEGs were upregulated in the B and C groups during passion fruit development (Fig 5a and 5c), which included certain key enzyme unigenes, such as CHS (Cluster-29126.6929), ANR (Cluster-29126.6305), and DFR (Cluster-29126.9769). At the same time, 11 DEGs were also identified that were involved in 'carotenoid biosynthesis' (ko00906), including CrtB (Cluster-29126.19339), CrtZ (Cluster-29126.16831), and NCED (Cluster-29126.17170) (Fig 5b). The expression levels of these unigenes in the B and C groups were more than 10-fold higher than those in the A group. Therefore, the results suggest that these unigenes may play an important role in regulating the coloration of passion fruit during its growth and ripening.

In addition, we also observed that certain DEGs were significantly enriched in 'phenylpropanoid biosynthesis' (ko00940) during the development of passion fruit. We all know that the

downstream pathway is the synthesis of lignin and anthocyanin (flavonoids). In this study, certain key regulated unigenes in 'phenylpropanoid biosynthesis' were identified as being significantly upregulated in stages B and C compared with A. Among these genes, CCR (Cluster-29126.13688) and F5H (Cluster-29126.6512) were upregulated significantly, which is involved in the regulated degree of lignification in passion fruit during its development (Fig 5c). Thus, we inferred that the degree of lignification was also related to ripening or senescence in passion fruit.

## KEGG analysis of DEGs in postharvest storage

Dynamic processes of passion fruit ripening and senescence were identified by enriched KEGG pathways of the control, 1-MCP and PF treatments. For 8D of storage (control), the DEGs of E vs C were significantly enriched in 'starch and sucrose metabolism' (ko00500), 'plant hormone signal transduction' (ko04075), 'phenylpropanoid biosynthesis' (ko00940) and 'flavonoid biosynthesis' (ko00941). For the 1-MCP- and PF-treated groups, the DEGs of G vs E and J vs E were also significantly enriched in 'starch and sucrose metabolism', 'plant hormone signal transduction', 'phenylpropanoid biosynthesis' and 'flavonoid biosynthesis' (Fig 6). These results indicated that 'phenylpropanoid biosynthesis', 'starch and sucrose metabolism' and 'plant hormone signal transduction' play vital roles in regulating fruit senescence. Then, in the KEGG pathway analysis, unigenes involved in 'starch and sucrose metabolism' and 'plant hormone signal transduction' were further analyzed.

Cell wall metabolism and signal transduction pathways are involved in senescence of passion fruit. Most of the differentially expressed genes (DEGs) were enriched in cell wall metabolism-related pathways (SF2 and SF3 in S1 File; S1 Table), including sucrose synthase, cellulose synthesis, pectin metabolism (PG), hemicellulose metabolism (β-Gal), and lignin metabolism (PAL, 4CL, CHS, CCR and POD). In the 'starch and sucrose metabolism' (ko00500) pathway, the expression levels of cluster-29126.797, cluster-29126.15857, cluster-29126.19858 and cluster-29126.23754 el al. were significantly upregulated at the mature stage (both the B and C groups). However, these DEGs were all downregulated in the G and J groups by 1-MCP and PF treatment, respectively (Fig 7a). We found that the expression levels of these DEGs in the G group (PF treatment for 8D) were notably higher than those in the J group (1-MCP treatment for 8D). The DEGs in the 'starch and sucrose metabolism' pathway included E3.2.1.21 (beta-

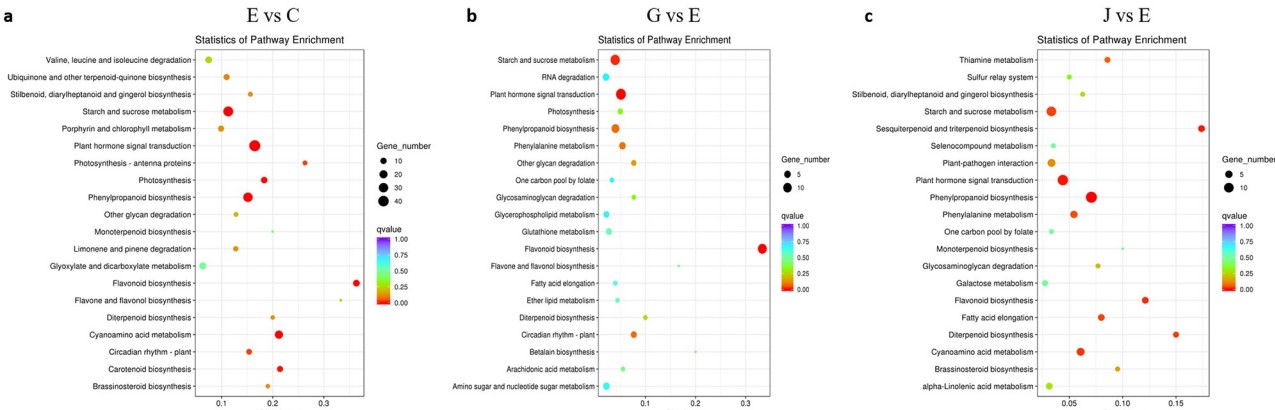

**Fig 6. Top 20 KEGG enrichment pathways in passion fruit during postharvest stage.** (a) The DEGs enriched KEGG pathway in E vs C. (b) The DEGs enriched KEGG pathway in G vs E. (c) The DEGs enriched KEGG pathway in J vs E. C (8 weeks post anthesis—physiological maturity), E (store at room temperature for 8 days), G (PF treatment for 8 days), and J (1-MCP treatment for 8 days). Rich factor was represent the degree of DEGs enrichment in this pathway.

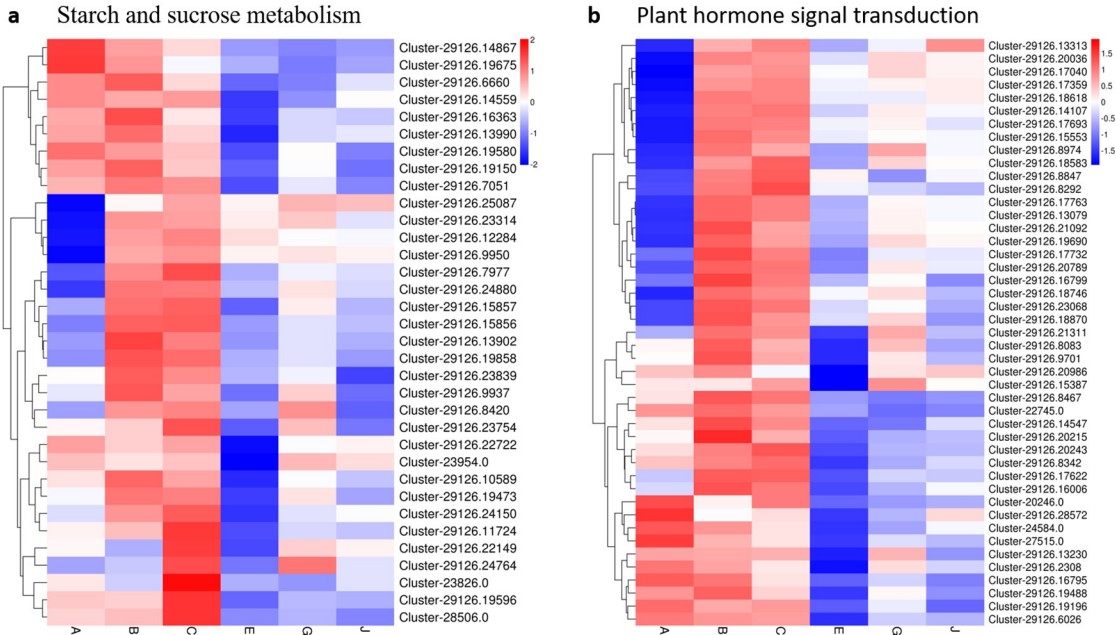

**Fig 7. The heat-map of DEGs expression profile related to starch and sucrose metabolism (a) and plant hormone signal transduction (b) in passion fruit during postharvest storage with different treatment. Red indicates up-regulated and blue was down-regulated DEGs expression. A (4 weeks post anthesis), B (6 weeks post anthesis), C (8 weeks post anthesis—physiological maturity), E (store at room temperature for 8 days), G (PF treatment for 8 days), and J (1-MCP treatment for 8 days).** The corresponding index are listed in SF3 and SF4 in S1 File. Data are presented as the mean ± SD of three biological replicates.

glucosidase), E2.4.1.14 (sucrose-phosphate synthase), E3.1.1.11 (pectinesterase) and amyA (alpha-amylase).

Similar to the 'starch and sucrose metabolism' pathway, the expression levels of DEGs enriched in 'plant hormone signal transduction' (ko04075) were analyzed in this study. A total of 45 DEGs of the 'plant hormone signal transduction' pathway was upregulated during passion fruit ripening, while these DEGs were downregulated in the process of senescence (Fig 7b). We found that the expression levels of these DEGs in the PF- and 1-MCP-treated (G and J) groups were upregulated compared with those in the control (E) group at 8D. The DEGs were key unigenes that regulated ethylene and auxin synthesis, including certain important enzyme unigenes, such as ETR, EBF1_2, and CTR1. These results indicated that cell wall metabolism and the plant hormone signal transduction (ethylene and auxin) pathway were essential to regulate passion fruit ripening and senescence. We also found that PF and 1-MCP treatment can affect the expression of these cell wall metabolism-related unigenes to delay the senescence of passion fruit.

### Transcriptome validation through qRT-PCR

To validate the transcriptome sequencing data, eighteen unigenes with different expression patterns were selected for qRT-PCR analysis (S2 Fig). These genes included three 'carotenoid biosynthesis'-related genes (Cluster-29126.19339, Cluster-29126.16831, Cluster-29126.17170); four 'flavonoid biosynthesis'-related genes (Cluster-29126.6929, Cluster-29126.23818, Cluster-29126.9769, Cluster-29126.22876); three 'phenylpropanoid biosynthesis'-related genes (Cluster-29126.13688, Cluster-29126.5571, Cluster-29126.6512); three 'plant hormone signal transduction'-related genes (Cluster-29126.17622, Cluster-29126.21896, Cluster-29126.8974); two

'starch and sucrose metabolism'-related genes (Cluster-29126.10589, Cluster-29126.16363); and three 'cell wall synthesis and degradation'-related genes (Cluster-29126.23818, Cluster-29126.22988, Cluster-29126.10249). These genes are involved in signal transduction, coloration, cell wall function, respiration and energy, and other types of metabolism. The results of RNA sequencing and qRT-PCR data showed a positive correlation in this study ($R^2 = 0.564$, $p < 0.01$, S2 Fig). This result indicated that the RNA-Seq data were reliable for further analysis.

## Discussion

### Changes in pericarp color and related molecular mechanisms during ripening

The pericarp color changed from green to purple in passion fruit during the ripening stage. In stages A, B and C of passion fruit, the expression patterns of DEGs were significantly enriched in 'phenylpropanoid biosynthesis', 'flavonoid biosynthesis and 'carotenoid biosynthesis'. A large number of studies have reported that 'flavonoid biosynthesis', 'carotenoid biosynthesis', and genes downstream of the 'phenylpropanoid biosynthesis' pathway are involved in the coloration of fruit [27, 28] by anthocyanin and carotenoid synthesis and accumulation. The anthocyanin and carotenoid contents contribute to the color of the fruit and may contribute to human health-promoting properties.

### Changes in the physiological index and related enzymes during postharvest storage

Climacteric fruit usually suffers from rapid senescence after harvest and exhibits an increase in weight loss, respiratory rate, ethylene production rate and physiological disorders at room temperature storage. The effects lead to fruit quality deterioration, tissue softening, dehydration and flavor changes [29, 30]. Purple passion fruit exhibits a typical climacteric pattern during postharvest ripening [31]. Therefore, to study the mechanism underlying fruit senescence and preservation during the storage process, we determined the physiological and biochemical indexes in passion fruit by PF and 1-MCP treatment at room temperature storage. First, weight loss and respiratory rate were determined at different storage time points with PF or 1-MCP treatment. Respiratory burst was elicited through aerobic and anaerobic respiration, which was regulated specifically by related signal transduction. In this study, weight loss and respiratory rate increased progressively and peaked at the later stage of storage at room temperature. However, this increasing trend was significantly inhibited by 1-MCP and PF treatment, and the lowest weight loss and respiratory rate were shown in PF-treated fruit. The respiratory burst was mainly related to increased production of reactive oxygen species (ROS), enhanced peroxidase activity and stimulation of the lipoxygenase pathway [32], which were closely related to accelerated senescence of fruit.

The ROS accumulation level was controlled by the balance between the capacity of ROS production and scavenging [33]. The cell membrane maintains relative stability in the internal environment and might be damaged by excess ROS levels, which also cause peroxidation and accelerate senescence in fruit [34]. Therefore, enzymatic and nonenzymatic ROS scavenging systems reduce potential damage to cells [35, 36]. The antioxidant enzymes of ROS scavengers include superoxide dismutase (SOD), peroxidase (POD), and catalase (CAT) [37]. Our results indicated that applying 1-MCP and PF treatments postharvest could enhance SOD, POD, and CAT activities in passion fruit, and the related gene expression pattern was also consistent

with the results of RNA-Seq analysis. In addition, fruit also contains a variety of antioxidants, including phenolic compounds and anthocyanins [38].

## Plant hormones involved in fruit senescence at postharvest storage

In addition to ROS, other signaling pathways or molecules are also involved in plant senescence, including plant hormone transduction, ethylene, auxin, jasmonic acid and salicylic acid [39]. Plant hormones are indispensable in the regulation of fruit ripening and senescence, which control fruit color, sugar, flavor and aroma during ripening and senescence [40]. A previous study reported that ethylene and auxin play a major role in the ripening and senescence process of climacteric fruits [41]. In the present study, the results demonstrated that a large number of DEGs were enriched in plant hormone signal transduction and upregulated in ethylene and auxin synthesis pathways during fruit ripening (B and C), while they were significantly downregulated during the senescence process at postharvest storage. These DEGs include the ethylene and auxin synthesis-regulated key receptor unigenes ETR, EBF1_2, and CTR1. PF and 1-MCP treatment can mitigate the low expression levels of these DEGs in passion fruit during postharvest storage.

## Membrane and cell wall component changes during postharvest storage

The process of fruit senescence after harvest leads to the irreversible destruction of membrane integrity and to accelerated leakage of ions [42]. Loss of membrane integrity may lead to subcellular decompartmentalization, which results in enzymatic browning catalyzed by peroxidase and polyphenol oxidase in postharvest fruits [43]. The membrane integrity damage is reflected by membrane permeability, which can be determined by the relative electrical conductivity (REC) [44]. In the present study, REC increased under 1-MCP treatment and control conditions after storage, while PF treatment notably repressed the increases in REC parameters and regulated the gene expression of the 'biosynthesis of unsaturated fatty acids' pathway. This result suggests that the prevention of passion fruit senescence by PF treatment might be involved in decreasing the oxidative destruction of membranes. Consistent with our results, a previous study reported that melatonin inhibited membrane phospholipid degradation and maintained the degree of unsaturated fatty acids, which contributed to the preservation of membrane integrity in tomato fruit [45]. The degradation of unsaturated fatty acids results in destruction of the cell membrane integrity of the fruit peel [46, 47]. PF and 1-MCP treatment inhibited membrane lipid peroxidation, which was beneficial for maintaining unsaturated degradation to saturated fatty acids, which can maintain passion fruit cell membrane integrity.

Cell wall degradation is essential for fruit quality during fruit ripening and the senescence process [40]. The thickness and strength of the cell wall are key components for maintaining fruit firmness [48]. In the process of fruit senescence, the progressive disassembly of the primary cell wall structure and components is depolymerized by the action of cell wall hydrolases [49]. Senescence-related degradation of cell wall composition enzymes primarily includes β-galactosidase (β-Gal), polygalacturonase (PG), pectin methylesterase (PME), pectin lyase (PL), cellulase and expansin [50, 51]. In the present study, certain cell wall metabolism-related pathways were identified by RNA-Seq data, such as cellulose synthesis, pectin metabolism, and hemicellulose metabolism, and most of the DEGs were significantly upregulated during fruit ripening. In keeping with the RNA-Seq data, the activities of β-Gal, PG and cellulase were also significantly increased in passion fruit during storage, which were all inhibited by PF and 1-MCP treatment. Under PF treatment, the activity of these three enzymes was lower than that of 1-MCP and the control group. PG is regarded as a pectin-degrading enzyme in fruit [52], leading to cell wall loss and fruit softening [53]. β-Gal is a pectin-debranching enzyme that is

capable of simultaneously modifying pectin and hemicellulose [54]. Cellulase is widely regarded to cause the degradation of cellulose matrix in the cell walls of fruit [55]. In the present results, the integrity of the cell wall was well maintained by PF and 1-MCP treatment to inhibit fruit softening, and PF treatment showed better results than 1-MCP. Therefore, our results suggest that PF treatment can alleviate passion fruit quality deterioration and suppress the activities of β-Gal, PG and cellulase to inhibit cell wall component degradation.

### Expression of sucrose and cell wall degradation-related DEGs during postharvest storage

The contents of sugar, fructose and glucose are the key factors in the formation of fruit quality and accumulate significantly during the fruit ripening process [56, 57]. However, the sucrose and fructose content of passion fruit decreased during postharvest storage [7]. In this study, the DEG expression levels of the starch and sucrose metabolism pathway were significantly upregulated at the ripening stage in passion fruit, while these DEGs were all downregulated in the G and J groups by 1-MCP and PF treatment. Therefore, we speculate that PF and 1-MCP can delay the senescence of passion fruit.

Starch, hemicelluloses, cellulose, and pectin are the major factors of cell wall polysaccharides [58]. In the present study, these related pathways, including sucrose metabolism and lignin metabolism, were identified by RNA-Seq. Most of these unigenes were involved in the cell wall polysaccharides that were significantly affected by PF and 1-MCP treatment. Lignin is one of the most abundant polyphenolic polymers in higher plants and functions in the structural support of the cell walls, water tightness, and response to environmental stimulation [59]. Several studies have reported that the activities of POD, PAL, C4H, and 4CL are positively correlated with lignin accumulation in loquat fruit [60]. In the present study, lignin metabolism-related unigenes in the 'phenylpropanoid biosynthesis' pathway, such as POD, PAL, C4H, and 4CL, were identified by RNA-Seq analysis,. The results of this study are consistent with prior findings, which observed that several postharvest treatments, such as 1-MCP and PF, could inhibit lignification in fruit [61].

In general, these results are consistent with the findings of previous studies showing that exogenous 1-MCP, melatonin or glycine betaine can delay postharvest senescence in fruits [47, 62] by reducing respiratory rates, REC, and key enzyme activities involved in cell wall degradation and by increasing antioxidant enzyme activities, such as the activities of β-Gal, PG, cellulase, SOD, POD and CAT, and related gene expression levels. PF treatment can slow the respiratory rate of passion fruit and consequently lower the deterioration of organoleptic traits, such as flavor, aroma and cell wall, among other quality characteristics.

### Conclusions

To investigate the mechanism of ripening and epidermal senescence of *Passiflora edulia* Sims, transcriptome analysis and phytochemical characteristic analysis were conducted in purple passion fruit at different stage and treatment (PF and 1-MCP). A total of 9811 DEGs were identified during ripening stage. KEGG enrichment analysis was performed that phenylpropanoid biosynthesis, carotenoid biosynthesis and flavonoid biosynthesis were the key pathways for coloration and quality formation of passion fruit during maturation. In addition, the analysis of DEGs were involved in starch and sucrose metabolism, plant hormone signal transduction and phenylpropanoid biosynthesis at the postharvest stage. The expressed gene and enzyme activities of SOD, CAT, POD and lignin accumulation were significantly increased, while β-Gal, PG and cellulose activities declined by PF and 1-MCP treatment to delay cell wall degradation during fruit senescence. Therefore, both 1-MCP and PF treatment could inhibit

passion fruit senescence at storage stage. The effect of PF treatment may be superior to 1-MCP treatment to delay passion fruit senescence in this study. Certain candidate unigenes involved in coloration and senescence were screened in this study, which could be offer referees for future breeding programs and production of *Passiflora edulia* Sims cultivars with enhanced nutritional quality and to delay senescence.

## Supporting information

**S1 Fig. Clustering analysis of samples and the intersection of differentially expressed unigenes (DEGs) during fruit ripening and postharvest stages.** (a) Principal component analysis of the RNA-seq profile at different developmental stages of passion fruit, and postharvest stages with different treatment. (b) The DEGs counts of different comparison groups, grey indicates up-regulated and blue represents down-regulated.
(TIF)

**S2 Fig. Correlation between qRT-PCR and RNA-seq for the 18 unigenes.** Each point represents a fold change value of expression level between the corresponding FPKM and RT-PCR in passion fruit.
(TIF)

**S1 Table. List of primers used by RT-PCR and RT-PCR and FPKM by RNA-seq data.**
(XLSX)

**S1 File.**
(XLSX)

## Author Contributions

**Conceptualization:** Ming Xin, Xuemei He.

**Data curation:** Changbao Li, Li Li, Jinfeng Sheng.

**Formal analysis:** Changbao Li, Ming Xin.

**Funding acquisition:** Changbao Li, Ming Xin, Jian Sun.

**Investigation:** Xuemei He.

**Methodology:** Changbao Li, Li Li, Xuemei He.

**Project administration:** Jian Sun.

**Resources:** Changbao Li, Ming Xin, Guoming Liu, Jiemin Li, Jian Sun.

**Writing – original draft:** Changbao Li.

**Writing – review & editing:** Ming Xin, Li Li, Xuemei He, Guoming Liu, Jiemin Li, Jinfeng Sheng, Jian Sun.

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
