## [Decision Letter · Decision Letter 0]

7 Aug 2020

PONE-D-20-20801

Transcriptome profiling reveals the mechanism of ripening and epidermal senescence in passionfruit (Passiflora edulia Sims)

PLOS ONE

Dear Dr. Xin,

Thank you for submitting your manuscript to PLOS ONE. After careful consideration, we feel that it has merit but does not fully meet PLOS ONE’s publication criteria as it currently stands. Therefore, we invite you to submit a revised version of the manuscript that addresses the points raised during the review process.

We look forward to receiving your revised manuscript.

Kind regards,

Mohammad Ansari

Academic Editor

PLOS ONE

Journal Requirements:

3. Thank you for including your funding statement; "no"

Reviewers' comments:

Reviewer's Responses to Questions

**Comments to the Author**

1. Is the manuscript technically sound, and do the data support the conclusions?

Reviewer #1: Yes

Reviewer #2: Yes

2. Has the statistical analysis been performed appropriately and rigorously? 

Reviewer #1: Yes

Reviewer #2: Yes

3. Have the authors made all data underlying the findings in their manuscript fully available?

Reviewer #1: Yes

Reviewer #2: Yes

4. Is the manuscript presented in an intelligible fashion and written in standard English?

Reviewer #1: No

Reviewer #2: Yes

5. Review Comments to the Author

Reviewer #1: Submitted work is very interesting and passion fruit is gaining importance of study due to its nutritive value. However, manuscript requires major revision in terms of presentation.

Please go for english editing as many times in manuscript clarity of statement are missing ( e.g. Lines 42-47; 56-59; 95-99; 101-103; 159-161 etc.)

Manuscript also requires a strong conclusion section which is absent here. Please add.

Figures are not clear and somewhere unreadable.

Reviewer #2: In the article entitled "Transcriptome profiling reveals the mechanism of ripening and epidermal senescence

in passionfruit (Passiflora edulia Sims)", authors have done a good piece of work. However, few questins are still remaining as mentioned below:

1: What is novel in this study? What is explained in the manuscript is already well known in several other studies with different fruits.

2: The expression of genes in KEGG classification (Fig 3B) doesnot match with Fig 4B. Explain.

3: Legend of Figure 5 is confusing. A, B and C is confusing between A,B and C of each panel. The given pathway is also not been mentioned in the legend

4: When the hormone response was not observed in KEGG classification (Fig 3B), then why it is studied in Fig 7B?

5: The manuscript lacks in correlation between different components, which can be improved in the revised one.

6. PLOS authors have the option to publish the peer review history of their article (what does this mean?). If published, this will include your full peer review and any attached files.

Reviewer #1: No

Reviewer #2: No

---

## [Author Response · Author response to Decision Letter 0]

20 Aug 2020

Dear reviewer 1,

Thank you very much for your patience to read our article and putting forward certain key suggestions for modification. We have made explanations for your questions and revised the insufficient parts of the article according to your suggestions. The following is our reply to your questions one by one.

1: What is novel in this study? What is explained in the manuscript is already well known in several other studies with different fruits.

Response: Thank you very much for your very professional questions. The novel in this article is the first systematic comparison and analysis of different storage periods of passion fruit from immature to mature period and 1-MCP and preservative film (PF) treatment after picking. Although 1-MCP treatments have been extensively studied in other fruits, they have rarely been compared with PF treatment, and our results show that PF treatment appears to be more cost-effective than 1-MCP treatment (further research is needed to verify this conclusion). At present, a growing number of people advocate green healthy food that without food chemical additives and chemical pesticides. This is one of the reasons why we will carry out further research. Thanks again for your patience. If you have any question, please contact us. We will deal with it seriously and revise it seriously.

2: The expression of genes in KEGG classification (Fig 3B) does not match with Fig 4B. Explain.

Response: Thank you very much for your attention and asking the key questions. First of all, we apologize for the careless of not clearly describing and explaining figure 3B and 4B in the manuscript. Figure 3B shows the KEGG classification of all genes detected by the transcriptome (these genes include differentially expressed genes and non-differentially expressed genes). Fig 4B shows the top 20 KEGG enrichment of differentially expressed genes in passion fruit during ripening stage. The KEGG classification in Fig 3B belongs to a large category, and each category contains several KEGG pathways of minor classification, namely the specific pathway shown in Fig 4B. Figure 3B contains Figure 4B, so it is possible that the result of figure 4B does not exactly same as Figure 3B. It is our mistake that makes you unable to understand this paper. Therefore, we have revised this manuscript and the legend of picture. Thanks again for your patience. If you have any question, please contact us. We will deal with it seriously and revise it seriously.

3: Legend of Figure 5 is confusing. A, B and C is confusing between A, B and C of each panel. The given pathway is also not been mentioned in the legend

Response: Thank you very much for your professional suggestions. To prevent confusion among readers, we changed the ‘ABCD’ in the figure to ‘abcd’ to distinguish the sample name from the figure and manuscript. The flavonoids and anthocyanin biosynthesis, carotenoids biosynthesis and phenylpropanoid biosynthesis pathway in Fig 5 were all from the results of Fig 4. The flavonoids and anthocyanin biosynthesis pathway were the downstream of phenylpropanoid biosynthesis pathway. I'm sorry that we failed to explain the logical relationship clearly, which caused you a misdirection. The legend of Fig 5 was also revised according to your suggestion. Thanks again for your patience. If you have any question, please contact us. We will deal with it seriously and revise it seriously.

4: When the hormone response was not observed in KEGG classification (Fig 3B), then why it is studied in Fig 7B?

Response: Thank you very much for your professional suggestions. Figure 3B shows the KEGG classification of all expressed genes (include differentially expressed genes and non-differentially expressed genes) detected by RNA-Seq. The signal transduction in Fig 3B is a large classification that containing the plant hormone signal transduction in Fig 7B. I'm sorry that our carelessness caused you to have doubts about the logic of the article. We have revised the manuscript according to your question. We hope you can be satisfied with the revised version this time. Thanks again for your patience. If you have any question, please contact us. We will deal with it seriously and revise it seriously.

5: The manuscript lacks in correlation between different components, which can be improved in the revised one.

Response: Thank you very much for your professional suggestions. We are very sorry for your misunderstanding of this article due to our carelessness. We have tried our best to revise this article to make the logic of the manuscript more meticulous and the language more coherent according to your advice. We sincerely hope that you will be satisfied with the revised version. Thanks again for your patience. If you have any question, please contact us. We will deal with it seriously and revise it seriously.

Dear reviewer 2,

Thank you very much for your patience to read our article and putting forward certain key suggestions for modification. We have made explanations for your questions and revised the insufficient parts of the article according to your suggestions. The following is our reply to your questions one by one.

1. Please go for english editing as many times in manuscript clarity of statement are missing ( e.g. Lines 42-47; 56-59; 95-99; 101-103; 159-161 etc.)

Response: Thank you very much for your suggestion. We submitted our manuscript to ‘American Journal Experts (AJE)’ to English editing. They were very professional to proofreading the language problems and obscure sentences in the manuscript. Please check it again. Thanks again for your patience. If you have any question, please contact us. We will deal with it seriously and revise it seriously.

2. Manuscript also requires a strong conclusion section which is absent here. Please add.

Response: Thank you very much for your professional suggestion. We have added the conclusion section at the end of the manuscript, please check. Thanks again for your patience. If you have any question, please contact us. We will deal with it seriously and revise it seriously.

3. Figures are not clear and somewhere unreadable.

Response: Thank you very much for your professional suggestion. We try our best to revised the Figures. And we hope that this revised version will not be ambiguous. Thanks again for your patience. If you have any question, please contact us. We will deal with it seriously and revise it seriously.

---

## [Decision Letter · Decision Letter 1]

14 Sep 2020

Transcriptome profiling helps to elucidate the mechanisms of ripening and epidermal senescence in passion fruit (Passiflora edulia Sims)

PONE-D-20-20801R1

Dear Dr. Xin,

We’re pleased to inform you that your manuscript has been judged scientifically suitable for publication and will be formally accepted for publication once it meets all outstanding technical requirements.

Kind regards,

Mohammad Ansari

Academic Editor

PLOS ONE

Additional Editor Comments (optional):

Reviewers' comments:

Reviewer's Responses to Questions

**Comments to the Author**

1. If the authors have adequately addressed your comments raised in a previous round of review and you feel that this manuscript is now acceptable for publication, you may indicate that here to bypass the “Comments to the Author” section, enter your conflict of interest statement in the “Confidential to Editor” section, and submit your "Accept" recommendation.

Reviewer #1: All comments have been addressed

Reviewer #2: All comments have been addressed

2. Is the manuscript technically sound, and do the data support the conclusions?

Reviewer #1: Yes

Reviewer #2: Yes

3. Has the statistical analysis been performed appropriately and rigorously? 

Reviewer #1: Yes

Reviewer #2: Yes

4. Have the authors made all data underlying the findings in their manuscript fully available?

Reviewer #1: Yes

Reviewer #2: Yes

5. Is the manuscript presented in an intelligible fashion and written in standard English?

Reviewer #1: Yes

Reviewer #2: Yes

6. Review Comments to the Author

Reviewer #1: (No Response)

Reviewer #2: The authors have replied to all the comments made by me and I am satisfied by the comments of the authors.

7. PLOS authors have the option to publish the peer review history of their article (what does this mean?). If published, this will include your full peer review and any attached files.

Reviewer #1: **Yes: **Dr. Amit Verma

Reviewer #2: No

---

## [Editor Report · Acceptance letter]

16 Sep 2020

PONE-D-20-20801R1

Transcriptome profiling helps to elucidate the mechanisms of ripening and epidermal senescence in passion fruit (*Passiflora edulia* Sims)

Dear Dr. Xin:

I'm pleased to inform you that your manuscript has been deemed suitable for publication in PLOS ONE. Congratulations! Your manuscript is now with our production department.

Kind regards,

on behalf of

Dr. Mohammad Ansari 

Academic Editor

PLOS ONE